# SEBS as an Effective Nucleating Agent for Polystyrene Foams

**DOI:** 10.3390/polym13213836

**Published:** 2021-11-06

**Authors:** Alberto Ballesteros, Ester Laguna-Gutiérrez, Miguel Ángel Rodríguez-Pérez

**Affiliations:** 1Cellular Materials Laboratory (CellMat), Condensed Matter Physics Department, University of Valladolid, Paseo de Belen 7, 47011 Valladolid, Spain; marrod@fmc.uva.es; 2CellMat Technologies S.L., Paseo de Belen 9-A, UVA Science Park Building, 47011 Valladolid, Spain; e.laguna@cellmattechnologies.com

**Keywords:** organic phases, SEBS, rheology, cellular materials, polystyrene

## Abstract

Different percentages of an elastomeric phase of styrene-ethylene-butylene-styrene (SEBS) were added to a polystyrene (PS) matrix to evaluate its nucleating effect in PS foams. It has been demonstrated that a minimum quantity of SEBS produces a high nucleation effect on the cellular materials that are produced. In particular, the results show that by adding 2% of SEBS, it is possible to reduce the cell size by 10 times while maintaining the density and open cell content of the foamed materials. The influence of this polymeric phase on the glass transition temperature (Tg) and the shear and extensional rheological properties has been studied to understand the foaming behavior. The results indicate a slight increase in the Tg and a decrease of the shear viscosity, extensional viscosity, and strain hardening coefficient as the percentage of SEBS increases. Consequently, an increase in the density and a deterioration of the cellular structure is detected for SEBS amounts higher than 3%.

## 1. Introduction

Elastomeric phases, such as thermoplastic elastomers (TPEs), which present the elastic behavior of rubbery materials and the re-processability of thermoplastic polymers, have aroused the interest of the scientific community in recent years [1]. Among all of the TPE that are available, tri-block copolymers have been widely used. Some examples of them are poly(methyl methacrylate)-poly(butyl acrylate)-poly(methyl methacrylate)copolymers, also known as MAM, styrene-butylene-styrene (SBS), and styrene-ethylene-butylene-styrene (SEBS). This last material presents high resistance to degradation, which makes it interesting for blending with common thermoplastic materials such as polypropylene (PP) or polystyrene (PS) [2]. The incorporation of this organic phase in common thermoplastic polymers has been reported by several researchers with successful results when modifying the impact properties of the solid polymer matrix. For instance, Sang et al. reported that by adding 13 wt.% of SEBS to a PS matrix, it was possible to improve the impact strength up to 4.4 times compared to that of pure PS [3]. Furthermore, Lindsey et al. showed that the inclusion of 20% SEBS in blends of high-density polyethylene (HDPE) with PS increases the impact behavior of the blends by 100%, with the drawback of reducing the tensile strength and the elastic modulus [4]. Finally, Banerjee et al. reported that it was possible to reinforce the mechanical properties of a SEBS matrix by introducing a certain amount of PS. They also reported that the rheological behavior was modified. An increase in the PS content resulted in the lowering of the shear viscosity and energy requirements for mixing, indicating an easier flow and more sustainable processing [1].

However, the use of these elastomeric organic phases to improve the cell nucleation of common polymers during the foaming process has not been deeply studied. In general terms, the addition of a second phase creates interfaces in the polymer/gas mixture, and these surfaces induce wetting; that is, the gas molecules tend to aggregate at the foreign surface. Then, the nucleation process tends to take place on these pre-existing surfaces. This process is called heterogeneous nucleation. The nucleation rate (*N_HET_*) for heterogeneous nucleation is given by Equation (1) [5].
(1)$$N_{HET}=C_{1}f_{1}exp\left(\frac{-\Delta G_{het}^{\prime }}{KT}\right)$$
where *C*_1_ is the concentration of gas in the polymer, *f*_1_ is the frequency factor of the gas molecules, *k* is the Boltzmann constant, *T* is the temperature, and $\Delta G_{het}^{\prime }$ is the free energy barrier for the heterogeneous nucleation that should be surpassed to obtain a stable nucleus.

The paucity of the literature focused on studying these effects in the foaming mechanisms that are associated with the incorporation of a TPE phase to a thermoplastic polymer matrix could be due to the difficulties associated with the selection of proper organic phases. There are two key aspects that must be considered why trying to improve heterogenous nucleation and therefore of the foaming behavior. On the one hand, the elastomeric phase must be properly dispersed in the polymer matrix, and on the other hand, the elastomeric phase must have a high affinity with the blowing agent used during the foaming process [6]. It is well known that dispersion plays a key role in determining the final properties of the cellular materials. In fact, poor dispersion could deteriorate the final properties of the cellular materials to values even lower than those obtained with the virgin polymers [7]. The main advantage of using secondary organic phases such as TPE to promote nucleation is that it is possible to design systems that “self-assemble” during melt blending, creating a morphology with an excellent dispersion of very small domains in the secondary phase. However, this only can be achieved when using materials with proper chemistry between the phases and an adequate viscosity.

Some examples in the literature that have focused on using organic phases in cellular materials have been found. MAM or thermoplastic polyurethane (TPU) have been used successfully as nucleating agents in poly (methyl methacrylate) (PMMA), while polydimethylsiloxane (PDMS) has been used in PP and in PS. Bernardo et al. have demonstrated that by adding 10 wt.% of a MAM copolymer to a PMMA matrix, it was possible to reduce the cell size to 93% with respect to pure PMMA [8]. Bernardo et al. have also demonstrated that the addition of 2 wt.% of TPU allows the cell size of PMMA foams to reduce by 80% [9]. Haurat et al. reported the results obtained using two different additive core–shell (CS) particles based on PBA (poly (butyl acrylate)) and PMMA and MAM in a PMMA polymer matrix. The results indicated that a liquid-core CS presents advantages for a decrease in density, even at room temperature foaming. On the other hand, in a PMMA/20 wt.% MAM blend, through a quasi-one-step batch foaming, a “porous to nonporous” transition was observed on thick samples. Such a sharp porosity gradient (from nonporous transparent areas to porous opaque areas within the same sample) revealed a lower pore size limit of around 50 nm in a classical batch process in “mild conditions” [10]. Qingfeng et al. reported a remarkable increase in the cell density of PP foams with the inclusion of a 5.2 wt.% of PDMS [5]. Quiang et al. demonstrated that with the inclusion of just a 1 wt.% of PDMS to a PS matrix, it was possible to double the solubility of with respect to the CO_2_ obtained in the pure PS matrix [11]. Banerjee et al. studied the use of PS (10 wt.%, 30 wt.% and 50 wt.%) as a nucleation agent for the SEBS polymer. They observed an increase in the complex viscosity of the solid composites as the PS content increased. They also reported that when the materials were foamed at temperatures that were close to the glass transition (Tg) temperature of the PS, the rheological characteristics of the material controlled the expansion ratio and the shrinkage. On the other hand, when the foaming temperatures were lower than the Tg of the PS but were higher than those for the ethylene-butylene phase, PS acted as a nucleation agent for the SEBS material. In this situation, a 30 wt.% of PS led to a reduction in the cell size of 60% with respect to the pure SEBS. Higher contents of PS (50 wt.%) generate a co-continuous phase in the material, resulting in an increase in the cell size [12]. Finally, Sharudin et al. have also reported the effects of shear dynamic rheology and foaming behavior associated with the addition of PS to a SEBS matrix. They found that an increase in the styrene content led to an increase in the storage modulus and to a decrease in the gas permeability. As a result, the shrinkage of the foam was controlled, and stable microcellular elastomer foams were obtained [13].

After the literature search and as far as the author knows, there are not any works that analyze the use of SEBS as a nucleation agent in a cellular material based on PS.

Taking all of the previous ideas into account, in this work, different contents of SEBS, varying between 0.25 wt.% and 10 wt.%, were added to a PS matrix with the objective of improving the cell nucleation mechanisms. A significant reduction in the cell size can be only achieved if the system presents a proper dispersion of the elastomeric phases (ethylene-butylene) and if there a good interaction with the gas used as blowing agent (CO_2_).

The reduction of the cell size due to the improvement of the cell nucleation mechanisms and the improvement of the homogeneity of the cellular structure could have positive effects in the physical properties of the foamed samples by reducing the thermal conductivity and by improving the mechanical properties [14]. This is a very interesting result, considering that one of the main applications of PS foams is use as thermal insulators in the construction sector [15]. Furthermore, this work is the first one, as far as the authors know, that analyzes the effects of the SEBS as a cell-nucleating agent in PS foams.

## 2. Materials and Methods

A commercial polystyrene (PS) recommended for foam applications (Edistir N3840 from Versalis, San Donato Milanese, Italy) with a melt flow index of 10 g/10 min (200 °C/5 kg), a density of 1.05 g/cm^3^, and a glass transition temperature (Tg) of 89 °C was used as the polymer matrix. A commercial SEBS (Kraton G1643MS from Kraton Corporation, Houston, TX, USA) with a melt flow index of 17.6 g/10 min (200 °C/5 kg), a density of 0.91 g/cm^3^, and a styrene content of 20% was used as the secondary phase to produce the PS-SEBS blends. Before the materials were processed, they were dried in a vacuum drying oven (Mod. VacioTem TV, P-Selecta, Barcelona, Spain) at 70 °C for 4 h.

The mixing of the PS with the SEBS was conducted in a twin-screw extruder (Collin ZK 25 T with L/D of 24) following a temperature profile that increased from 145 °C to 185 °C (at the die) and with a screw rate of 50 rpm. Different formulations were produced by adding 0.25 wt.%, 0.5 wt.%, 1.5 wt.%, 3 wt.%, 5 wt.%, and 10 wt.% of SEBS to the PS matrix. After extrusion, the materials were pelletized, and then, the materials were thermoformed in a hot-cold press (at 235 °C and 27 bar), obtaining materials with the desired shape and size 2 × 2 × 0.2 cm (L × W × T) for the foaming experiments.

The glass transition temperature (Tg) of the pure PS and the materials containing SEBS was analyzed by differential scanning calorimetry (DSC) using a DSC 3 + from Mettler Toledo. A heating step from 20 °C to 160 °C at a heating rate of 10 °C min^−1^ was considered in this experiment. The measurements were performed in a nitrogen atmosphere with a flux of 60 mL/min.

The shear dynamic rheology was measured using a stress-controlled rheometer (AR 2000 EX from TA Instruments, New Castle, DE, USA). Dynamic shear measurements were performed at a temperature of 220 °C under a nitrogen atmosphere and using parallel plates with a diameter of 25 mm. A fixed gap of 1 mm was selected to perform the rheological measurements. First, a strain sweep test, at a fixed dynamic frequency (1 rad s^−1^), was performed to determine the linear viscoelastic regime of the different blends. It was found that the employed strain should be 3%. Later, a time sweep was performed to recover the initial state of the material, which was partially deformed when the sample was loaded in the rheometer. The duration of the time sweep varied between 360 and 600 s, depending on the material. Finally, the frequency sweep step was performed in a range of angular frequencies varying between 0.01 and 100 rad s^−1^. From these measurements, four parameters were analyzed: the dynamic shear viscosity in the terminal region, also known as zero shear viscosity (|*η**|), the slopes of the storage and loss modulus (*G*′ and *G*″) in the terminal region, and the cross over points among the two curves (*G*′ and *G*″). The same rheometer but with an extensional fixture (SER 2 from Xpansion Instruments, Tallmadge, OH, USA) was used to analyze the extensional rheological behavior of the different formulations. In this device, the samples were clamped to two cylinders that rotate in opposite directions at a fixed rate while applying a uniaxial stretching force to the material. All of the experiments were conducted at a temperature of 160 °C and at different Hencky strain rates: 0.3, 0.5, and 1 s^−1^. In all the experiments the maximum Hencky strain was 2.8. From these experiments, the extensional viscosity was obtained as the ratio between the measured stress and the corresponding Hencky strain rate. A more detailed description of the measurement protocol can be found elsewhere [7]. From the extensional viscosity measurements, the strain hardening coefficient (S) was obtained. This parameter (see Equation (2)), which allows the way in which the extensional viscosity increases when time or strain increase to be quantified, has been obtained for the different formulations.
(2)$$S=\eta_{E}^{+}(t,\dot{\varepsilon_{0})}/\eta_{E0}^{+}\left(t\right)$$
where ($\eta_{E}^{+}(t,\dot{\varepsilon_{0})}$) is the transient extensional viscosity for a determined time (t) and Hencky strain rate ($\dot{\varepsilon_{0}}$), and $\eta_{E0}^{+}\left(t\right)$ is the transient extensional viscosity in the linear viscoelastic regime, which can be obtained in two different ways: as three times the time-dependent shear viscosity growth curve at very low shear rates or by extrapolating the overlapping parts of the extensional curves at different elongation rates [16]. In the present work, the second option was chosen to obtain the strain-hardening coefficient. This coefficient was determined for a time of 2.67 s and for a Hencky strain rate of 1 s^−1^.

The morphology of the solid (non-foamed) PS-SEBS blends was analyzed by scanning electron microscopy (SEM). First, the materials were frozen in liquid nitrogen and were fractured afterwards. The fracture surface was made conductively by means of the sputtering deposition of a thin layer of gold, and later, SEM micrographs were obtained by using a FlexSEM 1000 from Hitachi (Hitachi, Japan).

Foams were produced using the gas dissolution foaming process using a saturation pressure of 8 MPa and a saturation temperature of 40 °C for 8 h. CO_2_ was used as the blowing agent [17]. Once the samples were removed from the pressure vessel, they were introduced in a thermostatic oil silicone bath for 1 min at 120 °C to produce the expansion of the different materials.

Before characterizing the cellular materials, the solid skin of the foamed samples was removed using a polishing machine model LaboPOl2-LaboForce3 from Struers (Cleveland, OH, USA). Then, the densities of the cellular materials without skin were determined (ASTM standard D1622-08). In addition, the open cell content was measured using a gas pycnometer according to the standard ASTM D6226 and using nitrogen as the gas for the measurement.

The cellular structure of the foamed samples was characterized by SEM using the same microscope and a similar procedure to prepare the samples to those employed with the solid materials. Parameters such as the average cell size (***Φ***), the cell nucleation density (*N*_0_), and the standard deviation of the cellular structure (SD) were measured using an image processing tool based on the software Fiji/Image J (with 64 bit, Java 1.8) [18].

## 3. Results and Discussion

To improve the nucleation mechanisms during the foaming process with the aim of improving the cell nucleation density and reducing the cell size, the dispersion of the secondary phase (SEBS) in the PS matrix should be as good as possible. Thus, an important step in this work was to determine if the elastomeric phase is well dispersed in the polymer matrix. The dispersion degree was analyzed qualitatively through scanning electronic microscopy.

SEM images of the solid (non-foamed) composites are shown in Figure 1. In these micrographs, it is possible to detect two different phases, the PS matrix and the SEBS domains. The domains have a spherical shape, and they are properly dispersed in the PS matrix, with diameters varying between 0.2 and 0.5 µm. Examples of the SEBS domains have been marked with circles in the SEM images to make them easier to see. The SEBS domains tend to agglomerate as the SEBS content increases. It is remarkable that for contents higher than 3 wt.%, the SEBS domains are bigger, as seen in the SEM micrographs. This fact could indicate that there is an optimal SEBS content that allows it to maximize its dispersion in the PS matrix.

Figure 2 and Table 1 show the results obtained after the DSC and the rheological characterization of the formulations containing SEBS. The results obtained by the DSC indicate that the Tg remains constant for SEBS contents lower than 1.5 wt.%. However, when the number of particles is higher than 3 wt.%, there is a slight increase in the Tg. For instance, the Tg of the samples containing the maximum amount of SEBS (10 wt.%) is 5 °C higher than that of the pure PS. On the other hand, the shear dynamic rheological results (Figure 2a) indicate that a decrease in the zero-shear viscosity is obtained when the amount of SEBS increases. This behavior is the expected when considering the significant differences between the melt flow indexes of the PS (10 g/10 min) and those of the SEBS copolymer (17.6 g/10 min), with the viscosity of the SEBS being smaller than that of the PS. The results also indicate that when the SEBS content is equal to or higher than 5%, the Newtonian regime is not detected, and at this frequency range, it is no longer possible to see a flat plateau, as seen in Figure 2a. Banerjee et al. reported that SEBS has a reduced complex viscosity compared to the one of PS and an increase in the value of the complex viscosity compared to the SEBS material, when different PS contents were introduced into the SEBS material [1]. Easy flowability is a common characteristic of TPE materials. When the SEBS content is not remarkable (lower than 1.5. wt.%), the decrement observed in the viscosity due the higher flowability of the material is not notorious. However, when the formulations present higher SEBS contents, the viscosity values decrease remarkably.

Table 1 also shows that all of the formulations present a single cross over point between the curves of the storage and the loss modulus. In the work of Banerjee et al., it is reported that when large domains of PS were created, the flowability of the materials changed due to the possibility of the SEBS being able to flow easily [1]. In the present work, we did not see a notorious change in the behavior of the viscosity with the increase of the SEBS phase, which could indicate that large domains of SEBS were not formed, even when adding a SEBS content of 10 wt.%. This fact is in agreement with the lack of change observed in the storage and loss modulus curves as well as in the cross over points between them.

The extensional rheological results (Table 1 and Figure 2b) show that the strain hardening coefficient of the materials is reduced as soon as the SEBS content increases. The extensional viscosity curves were multiplied by a factor (included in the figure) to make it possible to compare all of the materials in a single figure. The reduction in the strain hardening is especially remarkable for the formulations that contains high quantities of SEBS (higher than 5%). A reduction of the strain hardening coefficient could lead to worse foamability, resulting in materials that are not able to resist extension during the foaming step, leading to heterogenous cellular structures with large cell sizes and a high content of open cells [7].

Figure 3 shows the SEM images of the foams produced with the different formulations. Only a very small amount of SEBS (0.25 wt.%) is necessary to reduce the cell size from 90 μm to 20 μm (see Table 2). When the SEBS content is lower than 2 wt.%, it is possible to see that the cell size decreases as the cell content increases, while the foam density remains almost constant (varying between 31–32 kg/m^3^). This result indicates that the secondary phase (SEBS) acts as a very efficient cell nucleation agent. When the SEBS content varies between 2 wt.% and 5 wt.%, the cell size still decreases as the SEBS content increases. However, in these materials, an increase in the foam density can be detected as the SEBS content increases. Finally, for a SEBS content that is higher than 5 wt.%, a significant increase is detected in both the foam density and the cell size. The cell size of the material containing a 10 wt.% of SEBS (around 250 μm) is even higher than that of the foam produced with the pure PS.

Furthermore, the homogeneity of the cellular structure, which can be studied using the ratio between the standard deviation of the cell and the average cell size (SD/***Φ***), is lower in the foams containing a 10 wt.% of SEBS since higher (SD/***Φ***) ratio values were detected in these foams than in the rest of cellular materials.

These results cannot be explained by the differences observed in the Tg when ahigh contents of SEBS were added. When maintaining the foaming temperature, which was 120 °C for all of the materials in present study, an increase in the Tg indicates that the gap between the foaming temperature and the Tg is reduced. In other words, the polymer spends less time in a rubbery state, which is an optimum approach to reduce the degeneration mechanisms. The obtained results can be explained by considering the reduction of the strain hardening coefficient values that are observed in Table 1. If strain hardening is very low, then the polymer is not able to resist the elongational forces occurring during the foaming process, and it breaks. As a result, degeneration mechanisms such as coalescence are favored, leading to an increase in the foam density as well as to a deterioration of the cellular structure.

Apart from the already mentioned behavior in the density, the cell size and homogeneity of the cellular structure and the open cell content (see Table 2) increase slightly when SEBS particles are introduced during the formulation up to values of 3 wt.%. For higher SEBS contents, a significant increase is detected in this parameter, which can once again be explained by the reduction of the strain hardening coefficient.

Comparing the results obtained by using SEBS as nucleating agent with those obtained with other nucleating agents, such as talc or other nanoparticles such as sepiolites, it can be concluded that SEBS is the most efficient nucleating agent. A very small amount of SEBS (0.25 wt.%) can lead to a significant reduction in the cell size with respect to pure PS (approximately 78%). These reductions can only be achieved by incorporating high amounts of sepiolites (higher than 3 wt.%). Moreover, this significant reduction cannot be obtained with other nucleating agents such as talc [19].

## 4. Conclusions

In this paper, an elastomeric phase based on a SEBS copolymer was added to a PS matrix to evaluate its effect as a cell nucleating agent. The morphology of the blends consists of a properly dispersed elastomeric phase comprising spherical particles with sizes between 0.2 and 0.5 μm. The size of the dispersed domains increases when the amount of SEBS increases, and the rheological behavior of the formulations are significantly affected by the increase in the SEBS content. It was found that strain hardening is clearly reduced for SEBS contents above 5%. The SEBS phase showed a remarkable nucleation effect. By incorporating only a 0.25 wt.% of the elastomeric phase, it is possible to reduce the cell size by 5 times while also maintaining the density and open cell content. An increase in the SEBS content up to 2 wt.% allows the cell size to reduce by 10 times, with a small increase in density and open cell content being observed as well. However, higher quantities of SEBS (between 5% and 10%) strongly increase the foam density and the open cell content. The present article constitutes, as far as the authors know, the first proof that SEBS is a very effective nucleating agent for PS foams. Furthermore, this elastomer is more efficient than other nucleating agents that are commonly used to improve the foamability of PS such as talc or sepiolites.

## Figures and Tables

**Figure 1 polymers-13-03836-f001:**
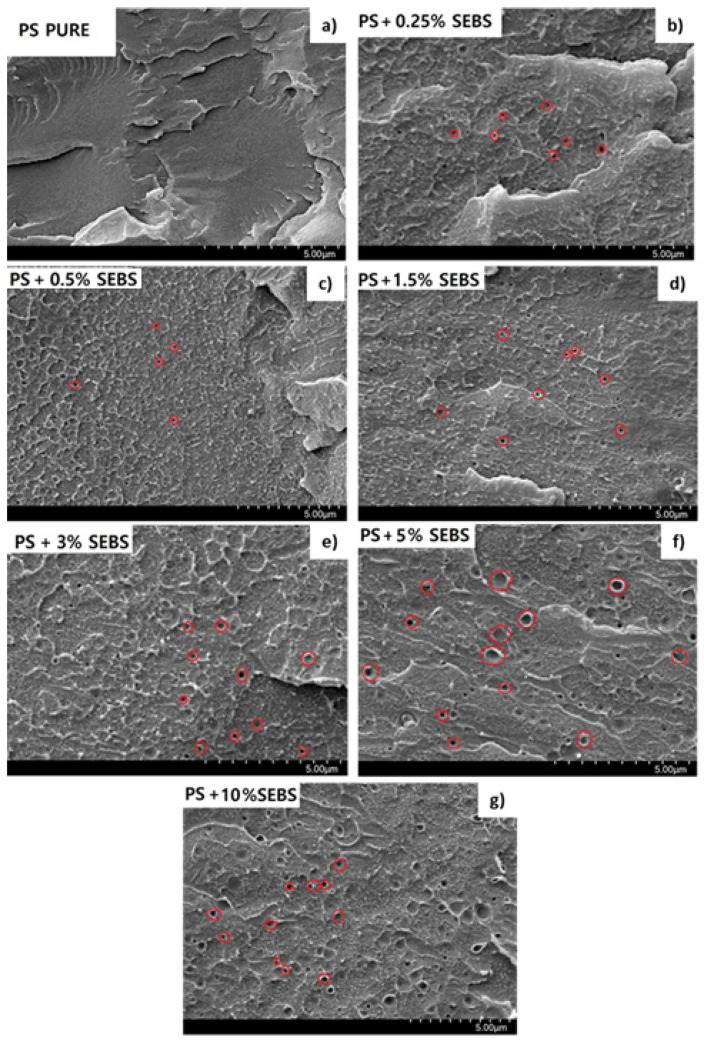
SEM micrographs of the solid formulations produced. (**a**) Pure PS; (**b**) PS + 0.25% SEBS; (**c**) PS + 0.5% SEBS; (**d**) PS + 1.5% SEBS; (**e**) PS + 3% SEBS; (**f**) PS + 5% SEBS; (**g**) PS + 10% SEBS. Red circles have been introduced to help with the visualization of the elastomeric phase.

**Figure 2 polymers-13-03836-f002:**
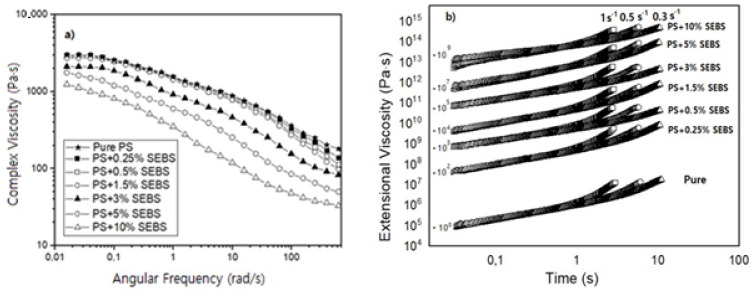
Results obtained after characterizing the rheological behavior of the different materials. (**a**) Shear dynamic rheology results. (**b**) Extensional rheology results.

**Figure 3 polymers-13-03836-f003:**
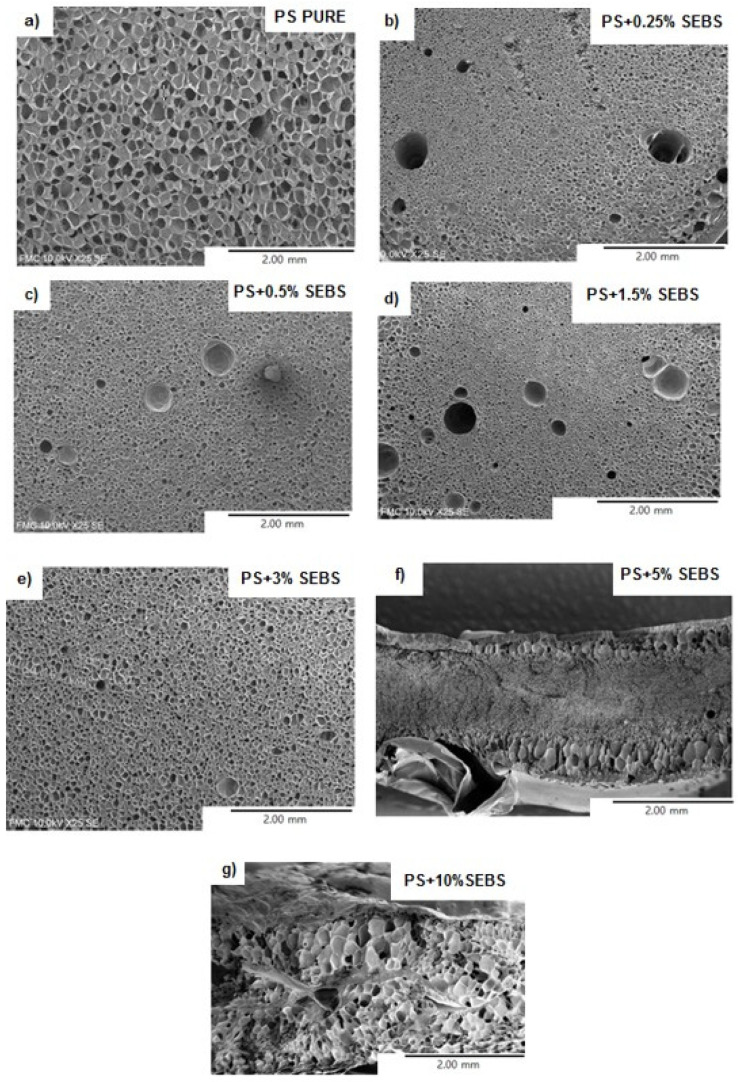
SEM micrographs of the cellular materials produced. (**a**) Pure PS; (**b**) PS + 0.25% SEBS; (**c**) PS + 0.5% SEBS; (**d**) PS + 1.5% SEBS; (**e**) PS + 3% SEBS; (**f**) PS + 5% SEBS; (**g**) PS + 10% SEBS.

**Table 1 polymers-13-03836-t001:** Glass transition temperature and shear and extensional rheology properties of the formulations produced.

Name of the Sample	Tg (°C)	Zero Shear Viscosity (Pa·s)	Slope of G′ (Pa·s)	Slope of G″ (Pa·s)	Cross over Points	Strain Hardening Coefficient
Pure PS	94.96	3000	1.84	0.98	0	3.83
PS + 0.25% SEBS	93.31	2987	1.80	0.98	1	3.81
PS + 0.5% SEBS	93.31	2904	1.76	0.96	1	3.78
PS + 1.5% SEBS	94.61	2742	1.72	0.94	1	3.55
PS + 3% SEBS	97.40	2123	1.63	0.87	1	3.26
PS + 5% SEBS	98.04	-	1.56	0.81	1	2.03
PS + 10% SEBS	100.37	-	1.23	0.72	1	1.46

**Table 2 polymers-13-03836-t002:** Density, cell size, cell nucleation density, homogeneity, and open cell of the foams produced.

Name of the Sample	Density (kg/m^3^)	Cell Size (µm)	Cell Nucleation Density (Nuclei/cm^3^)	SD/*Φ*	Open Cell Content
PS Pure	31.07 ± 0.67	90.02 ± 25.14	(4.76 ± 0.12) × 10^6^	0.27	12 ± 1.03
PS + 0.25% SEBS	31.15 ± 1.21	19.54 ± 3.14	(3.50 ± 0.25) × 10^8^	0.16	15 ± 0.63
PS + 0.5% SEBS	31.40 ± 0.78	16.40 ± 2.79	(3.76 ± 0.44) × 10^8^	0.17	13 ± 0.52
PS + 1.5% SEBS	32.16 ± 0.66	15.28 ± 2.18	(3.98 ± 0.63) × 10^8^	0.14	14 ± 0.47
PS + 3% SEBS	34.03 ± 1.03	8.14 ± 1.02	(4.36 ± 0.87) × 10^8^	0.12	17 ± 2.02
PS + 5% SEBS	41.62 ± 2.39	7.87 ± 2.33	(3.53 ± 0.51) × 10^9^	0.13	29 ± 3.74
PS + 10% SEBS	60.71 ± 4.56	256.3 ± 108	(9.63 ± 0.76) × 10^5^	0.42	45 ± 14.18

## Data Availability

Data are available upon request due to restrictions.

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
