# Peer review of "SEBS as an Effective Nucleating Agent for Polystyrene Foams"

_polymers, 2021, doi:10.3390/polym13213836_

Round 1

Reviewer 1 Report

It is recommended to accept after minor revision.

  1. The number 2 in CO2 is subscript, Please revise it in the Introduction.
  2. The number 3 of g/cm3 in line 77 on page 2 should be superscript.
  3. Is the unit of μmm in line 150 on page 4 correct?
  4. Please add the test method of open cell content.
  5. SEBS density is less than PS. After adding SEBS, the foam cell density increases and the open cell ratio increases. Why is the density not reduced?
  6. Supplement the conclusion section in accordance with the requirements of Polymers journal.
  7. Modify the results and discussion section of the manuscript. Because the foaming behavior is solid-state foaming, this is not a viscous-liquid polymer foaming process. Therefore, it is meaningless to use rheological data to explain the foaming behavior. Please modify the explanation in this part.

Reviewer 2 Report

This work describes the preparation of SEBS/polystyrene blends that were transformed into microcellular foams by a batch process using a gas as physical blowing agent. The authors employed electron microscopy to study the morphology of the polymer blends and the respective foams. To understand the nature of foam formation, rheology studies were performed.

The manuscript is deficiently written, which makes its evaluation difficult. Generally, this work is quite descriptive, the part dedicated to discussion of the results is too short and quite superficial. In its present form this submission is not suitable for publication and should be rejected. The authors may wish to consider the following recommendations for future submissions of this work.

  1. The Introduction is not coherent. The authors´ idea to consider polymer composites reinforced by inorganic particles together with porous polymers was not properly elaborated and in its preset form is quite confusing to the reader. The authors failed to explain the nature of the method they use for amorphous polymers´ foaming. It may be useful to include in the Intorduction the recent work of Haurat & Dumon in Molecules 2020, 25, 5320, as well as that of Banerjee, Ray & Gosh in J. of Cellular Plasticsc, Vol 53, Issue 4, 2017.    
  2. In the Introduction, the authors have to explain much better the novelty of their work; in the Conclusions the contribution of their studies to the present state-of-the-art should be clearly presented.
  3. The description of the sample preparation, especially that of the foams, should be made clearer. What was the gas applied? What were the reasons to choose the respective temperatures, times and pressures in this stage?
  4. From theoretical point of view the term “nucleation” is used mainly to describe the crystallization of polymers. If you want to apply it to bubble/pore formation due to depressurization of gases in amorphous polymer blends, a short explanation has to be presented.
  5. Figure 1, the insets to images a-f: a higher resolution must be used to show clearly the SEBS inclusions, instead of circling the morphological entities. The latter makes the insets unclear.
  6. The rheological data in Table 1 should be accompanied by a figure presenting some representative graphs from which this information was extracted.
  7. As stated before, the Conclusion part should be elaborated much better highlighting the utility of the results obtained in the present study.

Reviewer 3 Report

Different percentages of an elastomeric phase of styrene-ethylene-butylene-styrene (SEBS) were added to a polystyrene (PS) matrix to evaluate its nucleating effect in polystyrene (PS) foams. It has been demonstrated that some quantities of SEBS exhibits a high nucleation effect on the cellular materials produced.

The influence of this polymeric phase on the shear and extensional rheological properties have been studied to demonstrate foaming behaviour. The new information could be useful for polymer field researchers. The manuscript could be published after the revision.

*Authors describe in introduction of this manuscript nucleating agents for other polymers, for example: poly(methyl methacrylate) (PMMA), polydimethylsiloxane (PDMS) and others.  I would like to recommend a presentation here only the agents for polystyrene and to describe what is done in this field exactly with the PS.

*There are many nucleating agents for the PS, why the styrene-ethylene-butylene-styrene (SEBS) was chosen ?

* Could the authors explain why other simple  systems, for example as styrene-ethylene  or styrene-butylene are not investigated or not suitable as nucleating agents for the polystyrene matrix.

* I would recommend that the authors could present differential scanning calorimetry measurements for the prepared product in order to demonstrate thermal transitions in the final material.

* Influence of SEBS copolymer as nucleating agent for PS must be compared with that of other nucleating agents. It should be clearly demonstrated if this agent is better and the investigations can demonstrate or even discover a better product ?

Round 2

Reviewer 2 Report

The authors have dealt appropriately with the reviewer´s critical notes. The revised manuscript can be published. 

Author Response

Thank you very much for your suggestions and corrections.

Reviewer 3 Report

I recommend that the paper could be accepted after the final minor revision.

The authors should demonstrate DSC figures of the pure material and the materials that contains SEBS. It is an important and strange fact that Tg of both the materials are the same ?
